# Ovarian Cancer: Multi-Omics Data Integration

**DOI:** 10.3390/ijms26135961

**Published:** 2025-06-21

**Authors:** Anna Kliuchnikova, Arina Gordeeva, Aziz Abdurakhimov, Tatiana Materova, Svetlana Tarbeeva, Elizaveta Sarygina, Anna Kozlova, Olga Kiseleva, Elena Ponomarenko, Ekaterina Ilgisonis

**Affiliations:** Institute of Biomedical Chemistry, 119121 Moscow, Russia; a.kliuchnikova@gmail.com (A.K.); arina.atom@gmail.com (A.G.); az.abdurakhimov@gmail.com (A.A.); materova.ta20@physics.msu.ru (T.M.); tarbeevasn@gmail.com (S.T.); lizalesa@gmail.com (E.S.); ministreliya13113@gmail.com (A.K.); olly.kiseleva@gmail.com (O.K.); 2463731@gmail.com (E.P.)

**Keywords:** ovarian carcinoma, molecular heterogeneity, multi-omics, personalized therapy, biomarkers, omics

## Abstract

This study focuses on the systematization and integration of ovarian cancer multi-omics data, revealing patterns in the application of different omics-based approaches and assessing factors that affect the identification of potential biomarkers. An integrative analysis of 51 publications revealed 1649 potential biomarkers. The findings emphasized the molecular diversity of ovarian cancer. They demonstrated the importance of performing the comprehensive integration of molecular and clinical data to search for diagnostic alternatives and molecular patterns underlying ovarian cancer. The heterogeneity of data sources, differences in data acquisition and analysis protocols, and the lack of uniform standards affect the reproducibility of the results of genomic and post-genomic profiling. Multi-omics studies are more promising than mono-omics-based ones. Despite technological advances, researchers continue to focus on results obtained over a decade ago, which may hinder the scientific community from exploring new horizons in ovarian cancer research.

## 1. Introduction

The recent application of omics technologies, employing the integrative analysis of genomic, transcriptomic, epigenomic, proteomic, and metabolomic data, opens up new avenues for studying the biology of ovarian cancer. The integration of molecular information obtained from different omics levels is also known as multi-omics [1]. Not only does this comprehensive approach contribute to a deeper understanding of the mechanisms of tumor initiation and growth but it also enables the identification of novel therapeutic targets, biomarkers for early diagnosis, and the factors affecting sensitivity to therapy [2,3,4].

Ovarian carcinomas constitute a heterogeneous group of tumors with different molecular profiles. Cellular heterogeneity manifests itself as variations in gene expression, genetic mutations, and epigenetic alterations [5]. Epithelial ovarian cancer is the most common type of ovarian cancer, accounting for 90% of cases. Among epithelial tumors, serous carcinoma represents 52% of cases, particularly aggressive high-grade serous ovarian carcinoma (HGSC), which is often diagnosed at advanced stages (III–IV) [6]. This type of cancer is characterized by significant genomic instability (a tendency to accumulate a large number of genetic alterations such as mutations, chromosomal rearrangements, and copy number variations). It is more prone to DNA errors than other cells [5,7]. The key risk factors are mutations in *BRCA1/2* and *TP53* [8]. The endometrioid (10%), clear-cell (6%), and mucinous (6%, associated with smoking) subtypes are less common. Rare forms of carcinoma account for 26%. Non-epithelial ovarian cancers (10%) include germ cell and stromal tumors [6].

Omics studies, particularly multi-omics approaches, have gained considerable attention in oncology research, including ovarian cancer [9]. This trend has been driven by rapid advancements in next-generation sequencing, mass spectrometry, and computational methods for analyzing big data. According to databases such as PubMed and Web of Science, the number of multi-omics oncology publications is steadily growing. Despite the rapid development of multi-omics studies in oncology, the interpretation of their results remains challenging due to high methodological variability at each stage of analysis. Even when working with the same samples, differences in platforms and analytical approaches can lead to discrepancies in the results. For example, Ilgisonis et al. demonstrated that gene expression levels, as determined by RNA-Seq and qPCR, although correlated on average, exhibit significant discrepancies for individual transcripts, particularly in cases of low expression [10]. Likewise, Poverennaya et al. showed that shotgun proteomics and SRM (Selected Reaction Monitoring), when applied to the same samples, identified largely non-overlapping protein subsets due to differences in sensitivity and assay design [11]. Additional variation arises in plasma proteomics, where the use of depletion methods significantly affects protein detection. While depletion increases the likelihood of identifying low-abundance proteins, it simultaneously removes high-abundance diagnostic targets, limiting broader interpretation [12].

The development of computational algorithms and artificial intelligence, which enable the analysis of vast amounts of data and the unraveling of hidden patterns, is a key factor contributing to the growing popularity of multi-omics research. Furthermore, interdisciplinary collaborations between molecular biologists, clinicians, and bioinformaticians enable the integration of data from different omics levels (genomics, epigenomics, transcriptomics, proteomics, and metabolomics) to shape a holistic view of oncogenesis [13]. Not only does this approach contribute to more accurate patient stratification and personalized therapy selection but it also opens up new avenues in the development of diagnostic and prognostic biomarkers. For example, PINSPlus was applied to integrate DNA methylation and gene expression data, revealing novel subtypes of ovarian cancer with distinct survival outcomes and therapeutic vulnerabilities [14]. Similarly, CIMLR identified a high-risk subgroup in high-grade serous ovarian cancer (HGSOC) characterized by dysregulated immune pathways and poor prognosis, suggesting potential targets for immunotherapy [15]. The iCluF method further refined subtype classification by integrating copy number variation, methylation, and RNA-Seq data, leading to the discovery of a chemoresistant subgroup with elevated BRCA2 methylation and homologous recombination deficiency [16]. Additionally, Subtype-GAIN leveraged multi-omics data to impute missing molecular profiles and identified a stromal-rich subtype associated with extracellular matrix remodeling and worse overall survival [17]. These studies not only demonstrate the power of multi-omics integration in uncovering biologically and clinically relevant biomarkers but also highlight its potential to guide precision oncology strategies in ovarian cancer.

Multi-omics involves a large amount of experimental and bioinformatic data, which may potentially contain a previously unrevealed interplay between biological processes at different levels. Therefore, while the aggregate genomic, epigenomic, transcriptomic, proteomic, and metabolomic data can provide a better understanding of the molecular processes occurring in the body, the information noise present at each omics level may hinder the interpretation of the results. Although design and data handling standards have been established for some omics (e.g., genomics [18] and metabolomics [19]), there are currently no universal guidelines for multi-omics studies. Poor design of a multi-omics study may yield non-informative conclusions and lead to a lack of reproducibility. A good attempt to outline the recommendations has been reported in [20]. The guidelines for designing multi-omics studies in translational medicine, which enable the selection of omics data types and methods for their integration, are presented in this paper. Depending on the research objective, it is proposed to use different combinations of data. For detecting disease-associated molecular patterns, it is recommended to integrate the data that most fully represents pathogenesis. For identifying disease subtypes, it is advisable to combine the omics layers that show variability at the gene and protein expression levels. For diagnostic and prognostic purposes, one should utilize data that is closely related to phenotypic manifestations, such as proteomic and metabolomic data. When predicting the response to therapy, the focus should be on genomic variations and epigenetic alterations. Finally, for studying regulatory processes, it is crucial to use transcriptomics and epigenomics.

Recommendations for selecting data integration techniques based on specific objectives have also been formulated. Joint dimensionality reduction, factor analysis, and correlation methods are preferred for identifying key biological processes and associations with clinical features. Regression and graphical models are recommended for building regulatory networks and identifying relationships between the omics layers. It is reasonable to use deep learning techniques for integrating heterogeneous data and identifying complex nonlinear associations, especially when incorporating knowledge about biological networks to enhance the generalizability of models.

Despite the numerous mono- and multi-omics studies of ovarian cancer, only three FDA-approved diagnostic tests are currently used in clinical practice to assess the risk of this disease. The first one was the ROMA scoring system [21,22], which involves biomarkers *HE4* and *CA125* and is used to assess the risk of malignancy in adnexal masses in postmenopausal women. The next example is the OVA1 multi-marker panel [23,24], which comprises five biomarkers: β2 microglobulin (*B2M*), apolipoprotein A1 (*APOA1*), transthyretin (*TTR*), cancer antigen 125 (*CA125*), and transferrin (*TF*). This panel has been designed to assess the malignant potential of pelvic masses. Overa [25,26], the second generation of the OVA1 test, comprises two additional markers, *HE4* and the follicle-stimulating hormone. All these FDA-approved tests are characterized by sensitivities of up to 90% and specificities of 80%, as well as their features and limitations.

One of the key challenges in diagnostics is identifying universal biomarkers for the early detection of diseases. To achieve this objective, novel highly sensitive diagnostic methods, such as liquid biopsy [27] and nanosensor technologies [28], are being developed, and biomarkers are being studied at different omics levels [29], including metabolomics, microRNA expression profiles, and epigenetic markers. A multidisciplinary multi-omics approach combining these areas can significantly accelerate progress in disease diagnostics. Despite the growing popularity of the “multi-omics” term, it still lacks informative value for discoveries. Thus, quantitative measurements made by different laboratories and using different platforms are difficult to compare [30].

This paper aims to systematize and integrate all available data from academic publications focusing on the multi-omics of ovarian cancer. We strived to summarize the currently available qualitative data since even traits such as the presence of a mutation or upregulated gene expression can be usefully integrated into the multi-omics picture. This integration facilitates the detection of key patterns, the identification of common potential biomarkers, and the discovery of promising trends for further research.

## 2. Materials and Methods

A comprehensive list of potential biomarker protein molecules and their respective genes was obtained from the articles retrieved from the PubMed database [31] using the query “ovarian cancer multi-omics” over the past decade. The publications retrieved using the query “ovarian cancer + genomics OR transcriptomics OR proteomics OR metabolomics” do not fully coincide with those retrieved using the query “ovarian cancer multi-omics”. We believe that the reason for that lies in the original motivation of researchers to use one or several omics layers. A study published in 2011 [5] was added to the resulting list of publications as a good example of multi-omics data integration that has been cited in 38 publications analyzed by us. In this study, using the Cancer Genome Atlas (TCGA) project, we analyzed mRNA and microRNA expression, studied promoter methylation and the DNA copy number in 489 samples of high-grade serous ovarian adenocarcinoma, and examined the sequences of exons in protein-coding genes in 316 of these tumors [5].

Out of the 165 publications retrieved, we selected 51 papers (Appendix A) containing information on genes, transcripts, proteins, or their combinations that had been reported to have significant changes in the presence and/or quantitative content in patients with ovarian cancer (e.g., fold change in expression >2 and presence of a mutation or post-translational modification). The study selection criteria were registered in the PROSPERO system as a systematic review (PROSPERO registration number CRD420250614341) [32]. This study was conducted in compliance with the Preferred Reporting Items for Systematic Reviews and Meta-Analyses (PRISMA) statement [33]. Eight authors (A.K., E.S., S.T., A.I., A.A., A.K., T.M., and E.I.) independently conducted the study selection process; the data extraction process was divided equally between the researchers. In cases of disagreement regarding the inclusion of a specific study or during data extraction, an agreement was reached through discussion among all the authors. Not all the selected publications described “true multi-omics studies”. In some papers, the results of experiments conducted by different research groups were presented as review articles of multi-omics studies [9,34,35,36,37,38,39,40].

The K-Means clustering algorithm [41], followed by dimensionality reduction using principal component analysis (PCA) [42], was applied for the visual evaluation and classification of similar records. Coding for categorical variables was performed using OneHotEncoder [43]; the numerical data were normalized using the StandardScaler function [43]. Enrichment analysis was performed for each cluster using the KEGG database (version 2021; release 102.0, January 2021, pathway update 21 December 2020) and the Gene Ontology gene set (GO 2021-11-01 dump) utilizing the gseapy 0.13.0 library and the Enrichr tool (API 2023-08-01).

The DisGeNET platform (http://www.disgenet.org/, accessed on 7 March 2025, version 24.4) was used as an orthogonal approach to obtaining a list of biomacromolecules. To present the most comprehensive landscape of our current knowledge of the genetic underpinnings of human diseases, DisGeNET integrates data from expert-curated databases with information gathered through text-mining of the scientific literature [44]. In contrast to our primary method, which relies on manual data extraction from publications, the DisGeNET platform has yielded an independent, automatically aggregated list of disease-associated genes and proteins based on a large number of sources, including the results of genomic studies, association analyses, and annotated databases. This approach has ensured additional verification and expanded the list of molecules through systematic analysis of the knowledge gained.

The list of genes associated with ovarian cancer was retrieved using the DisGeNET platform under the query “ovarian cancer”. The query “ovarian cancer” comprised the following types of diseases: “Epithelial ovarian cancer, C0677886”, “Hereditary Breast and Ovarian Cancer Syndrome, C0677776”, “Familial ovarian cancer, C5679802”, “Carcinoma, Ovarian Epithelial, C4721610”, and “Malignant neoplasm of ovary, C1140680”. A comparative analysis with markers from the DisGeNET database was confined to records published within the same period (2014–2024) as our systematic review. To ensure high fidelity, only proteins with the “reviewed” status in UniProt were analyzed. Even the manually curated databases do not pay sufficient attention to cancer stages and types; therefore, our integration is important for improving the accuracy and contextual relevance of the lists of biomolecules. Thus, while analyzing the DisGeNET data, we have revealed cases where molecules were erroneously found to be associated with ovarian cancer, which may lead to the inclusion of non-specific or irrelevant markers. Our strategy of manual data extraction from original publications allowed us to correct these inaccuracies to end up with a more targeted set of biomarkers specific for high-grade serous ovarian cancer.

We used CancerMine (accessed on 30 May 2025) [45], a literature-mined resource of cancer-associated genes, to annotate candidate genes with their known roles as oncogenes, tumor suppressors, or cancer drivers. Only associations supported by curated evidence from peer-reviewed publications were considered. We downloaded the list of 938 genes associated with the “ovarian” type of cancer.

To further evaluate the relevance of identified genes to ovarian cancer, we queried the Open Targets Platform (accessed on 30 May 2025) [46]. This tool integrates diverse evidence types, including genetic associations, gene expression, somatic mutations, and known drug targets, to score gene-disease associations. A total of 9608 genes were associated with ovarian cancer, according to the Open Targets Platform.

To functionally characterize the genes identified only in our integrative analysis and lacking associations with cancer in other resources, we performed enrichment analysis using the Metascape platform (accessed on 30 May 2025, version 3.5.20250101) [47]. This is a web-based resource that integrates multiple bioinformatics databases and analytical tools. Gene Ontology (GO), KEGG pathway, Reactome pathway, and protein–protein interaction (PPI) network analyses were conducted by submitting gene lists to Metascape’s core analysis workflow with default parameters.

In particular, we used the Transcription Factor Targets enrichment module, which identifies the overrepresentation of known target genes of transcription factors based on data from the TRRUST and ENCODE databases. The analysis identifies transcription factors that may act as upstream regulators of the input gene set based on curated TF–target relationships from experimental evidence and literature mining.

Figure 1 illustrates the scheme for selecting publications for the systematization and integration of available multi-omics data for ovarian cancer.

After reviewing the selected papers, we identified 1649 potential biomarkers of various types, which were significantly different in the case of ovarian cancer (Appendix A). Appendix A describes each marker, including parameters such as the PubMed identifier (PMID), which allows for the prompt retrieval of the source, the marker name, the unique UniProt identifier, the gene name, and the full protein name. Moreover, the resulting list was annotated by key attributes, including the omics level (genomic, transcriptomic, or proteomic), the nature and direction of changes (e.g., down- or upregulated expression, presence of mutations, including SNPs, amplifications, and deletions, and various PTMs such as phosphorylation and hyper- or hypomethylation), the type of cancer associated with the marker, the number of samples examined, and the techniques used for biomarker analysis and verification. To provisionally indicate the significance of the publication in which a biomarker was identified, we added a column showing the number of citations of the paper, which is described in Appendix A. Hence, the identified potential biomarkers can be ranked according to their frequency of occurrence and the aforementioned characteristics.

## 3. Results and Discussion

### 3.1. The Current State of Multi-Omics Profiling of Ovarian Cancer Samples

This part of the study aimed to systematize the available multi-omics data obtained in ovarian cancer studies to identify recurrent patterns in the choice and application of different omics approaches, including transcriptomics, proteomics, genomics, and epigenomics. Particular focus was placed on analyzing methodological differences between studies, including the features of sample preparation, analysis platforms, and approaches to bioinformatics analysis. This comparative analysis allowed us to assess the extent to which technical and analytical differences affect the reproducibility and comparability of results, as well as the identification of potential biomarkers and molecular patterns associated with ovarian cancer.

To assess methodological diversity and focus areas in ovarian cancer research, we analyzed studies compiled in Appendix A. Figure 2 summarizes the distribution of studies by (a) analytical methods, (b) omics data types, and (c) ovarian cancer subtypes.

#### 3.1.1. Analytical Methods

We analyzed the data in Appendix A to assess the methodological diversity of multi-omics studies (namely, the types of samples and technologies used). The histogram in Figure 2a shows the distribution of analytical techniques used in multi-omics studies of ovarian cancer. The largest number of detected potential biomarkers (84.2%, *n* = 1518) relies on studies conducted using bioinformatics tools, thus emphasizing that in modern oncology, the focus shifts toward analyzing a priori datasets such as TCGA [48] and other open-access databases. In this context, “bioinformatics tools” refer specifically to studies based entirely on the secondary analysis of publicly available datasets without the generation of new experimental data. This classification does not exclude the use of bioinformatics in other methodological approaches but serves to distinguish purely computational workflows from those involving primary data acquisition. This approach enables the construction of predictive models, the identification of associations between clinical and molecular features, and the discovery of potential disease-associated molecules.

Sequencing, including RNA-Seq, ATAC-Seq, ChIP-Seq, and other next-generation sequencing techniques, is the second most frequently applied method (7.8%, *n* = 141, where *n* is the number of potential biomarkers of different types detected using this method).

Mass spectrometry was used in 3% of studies (*n* = 54) and encompasses the proteomic and metabolomic studies aimed at performing quantitative and qualitative assessments of proteins and metabolites. These methods are commonly used to validate disease-associated molecules at the level of proteins and metabolic products.

The integration of multi-omics data is used less frequently, in 2.2% of cases (*n* = 40). It is related to the high complexity of merging data from different levels (transcriptomics, proteomics, metabolomics, etc.); however, these approaches have the potential to provide a holistic view of the molecular heterogeneity of tumors.

Affinity methods, including immunohistochemistry, Western blot, and ELISA, were employed in 2.1% of studies (*n* = 38). Their role is usually to validate the results obtained using other methods, especially for clinical material.

Despite its high sensitivity and specificity, PCR was used in only 0.7% of cases (*n* = 12), indicating its limited application in the broad multi-omics context, especially in large samples.

Therefore, the data demonstrate that most multi-omics studies in ovarian cancer focus on the secondary analysis of publicly available (a priori) data using bioinformatics tools. This makes the studies reproducible and scalable but also emphasizes that the number of studies based on primary data needs to be increased to contribute to the discovery of new disease-associated molecules and clarify the molecular mechanisms of the disease.

#### 3.1.2. Distribution Across Omics Layers

The analysis of the distribution of studies according to the omics levels (Figure 2b) demonstrated that the greatest attention is paid to transcriptomic studies (60.1%), as confirmed by their high significance in the research into gene expression in various cancers, including ovarian cancer, due to the availability of data in open access databases. Genomic studies (20.5%) rank second, indicating an interest in analyzing mutations and epigenetic alterations. There are fewer proteomic studies (15.2%) despite their potential to identify disease-associated molecules. The “Other” category (4.1%) includes metabolomics, lipidomics, and complex multilayer analyses.

Proteomic studies are less common despite their importance in identifying protein biomarkers. Likewise, mass spectrometry remains a key tool in proteomics but is used less frequently compared to bioinformatics approaches. Despite their significance for identifying protein biomarkers, proteomic studies are relatively rare.

Transcriptomic studies are predominant due to the availability of RNA-Seq data in open-source repositories. Most studies are based on a priori data, as evidenced by the high proportion of bioinformatics methods. Sequencing, immunohistochemical, and mass spectrometry remain in demand but are used less frequently than a priori data analysis.

#### 3.1.3. Ovarian Cancer Subtypes

Next, we focused on the distribution of ovarian cancer types in the studied sample (Figure 2c). The pie chart illustrates the percentages of different ovarian cancer types in the context of omics research. The analysis revealed a marked predominance of high-grade serous carcinoma (HGSC) with a significantly lower representation of other tumor types. The distribution into categories is as follows:

*High-grade serous carcinoma (HGSC).* High-grade serous carcinoma (HGSC) is the predominant subtype (76.9%, *n* = 1457). This value is much higher than the prevalence of serous carcinoma reported in epidemiological studies [6]. HGSC accounted for 76.9% (*n* = 1457) of all cases, far exceeding its estimated population prevalence [6]. This reflects both clinical and methodological priorities. Clinically, HGSC is the most aggressive and lethal histological subtype, typically diagnosed at advanced stages (stage III—51%; stage IV—29%) and associated with poor prognosis (five-year survival of 42% and 26%, respectively). It also contributes disproportionately to ovarian cancer-related mortality. Methodologically, HGSC is comprehensively profiled in molecular consortia such as TCGA, making it the preferred model for secondary omics analyses. As a result, research interest is strongly concentrated on this subtype.

*Unspecified epithelial ovarian tumors.* Representing 21.5% (*n* = 410), these cases lack detailed histopathological or molecular classification. The absence of precise annotation reduces the resolution of subtype-specific analyses and may reflect limitations in the original clinical data sources.

*Low-grade serous carcinoma (LGSC).* This subtype was reported in only 0.5% (*n* = 9) of studies. Despite its recognized clinical and molecular distinctiveness, LGSC remains poorly characterized in high-throughput datasets, potentially due to its rarity and limited representation in consortia-scale projects.

*Clear cell ovarian carcinoma (CCOC).* Documented in only 0.2% (*n* = 3) of studies, CCOC appears markedly underrepresented compared to its reported population-level prevalence of approximately 6% [49]. This discrepancy may result from limitations in sample accessibility, exclusion criteria in study designs, or methodological difficulties in achieving standardized molecular profiling for this subtype.

*Borderline ovarian tumors.* Recorded in 0.4% (*n* = 7) of studies. Their limited representation suggests an absence of systematic molecular investigation, potentially attributable to their ambiguous classification or omission from conventional oncologic study designs.

*Mucinous ovarian cancer (MOC).* MOC was uncommon in the analyzed studies, accounting for 0.3% (*n* = 6), further indicating the rarity of this subtype.

Therefore, although HGSC is the predominant subtype, the presence of other histological subtypes attests to ovarian cancer heterogeneity, indicating that the subtypes need to be accurately identified in both molecular studies and clinical practice. Future efforts should prioritize standardized histological annotation and the systematic inclusion of underrepresented subtypes to capture the full molecular heterogeneity of ovarian cancer.

### 3.2. Integration of Multi-Omics Data and Their Functional Characterization

An analysis of 51 selected academic publications focusing on ovarian cancer omics revealed 1649 unique biological molecules that could be potential biomarkers (Appendix A). A total of 12 potential biomarkers (Figure 3a) identified in at least four datasets were key players in tumor pathogenesis, affecting the cell cycle, apoptosis, DNA repair, and the interplay with the immune system. It is worth noting that while HGSC is indeed the most prevalent subtype of ovarian cancer, this disproportionate focus significantly limits the generalizability of the findings to other histological subtypes.

Potential biomarker molecules mentioned in the largest number of publications were analyzed based on the literature. Mutations in *TP53* and its dysfunction reported in 11 papers were detected in 96% of the cases of highly differentiated serous ovarian cancer (HGSC), making it the major driver of tumor growth via the suppression of apoptosis and impaired cell cycle control [50]. Amplification of *CCNE1*, detected in six multi-omics studies, is associated with chemotherapy resistance and a poor prognosis, as it enhances cell proliferation through the hyperactivation of cyclin E1 [51,52].

The *BRCA1* and *BRCA2* genes, shown to be associated with ovarian cancer in at least five papers, are involved in double-stranded DNA break repair. Mutations in these genes increase the risk of ovarian cancer up to 40% and 10%, respectively, and are responsible for sensitivity to PARP inhibitors [53,54].

Mutations in the *KRAS* gene, reported to be a potential marker in five cases, activate the RAS/RAF/MEK/ERK signaling pathways, thus promoting tumor growth and resistance to therapy [55]. The loss of function of *RB1*, as reported in five publications, results in impaired cell cycle control and correlates with tumor aggressiveness. However, the simultaneous *RB1* loss and the BRCA-positive mutation status may improve survival due to increased sensitivity to chemotherapy [56,57].

According to [58], the amplification of *MYC*, identified as a potential biomarker in five cases, is found in almost 50% of HGSC cases and supports oncogenic cell growth, making MYC a promising therapeutic target. Increased expression of *EGFR*, as reported in four publications, is observed in approximately 50% of patients and is associated with an aggressive disease course and metastatic spread, likely via the activation of autocrine mechanisms [59]. *CCND1* and *VEGFA*, potential biomarkers reported in four publications, promote cell cycle progression and angiogenesis, respectively, thus contributing to tumor growth [51,52].

Proteins encoded by the *STAT1* and *MUC16* genes, as reported in four publications, are involved in the immune response and metastatic spread. *STAT1* regulates antitumor immunity, while *MUC16* (*CA125*) promotes tumor cell evasion from immune surveillance [53,60]. These proteins are components of complex signaling pathways and are responsible for tumor aggressiveness and the response to therapy.

### 3.3. Potential Ovarian Cancer Biomarkers: Literature and Resource Comparison

We compared the list of potential ovarian cancer biomarkers obtained from multi-omics studies with the list of proteins retrieved from the DisGeNET database, as well as from two well-established literature-mined resources of cancer-associated genes: CancerMine and the Open Targets Platform.

The DisGeNET database [61] is a platform containing information on the associations between genes and diseases. Biomarker–disease associations in the database can show both direct cause–effect relationships (e.g., driver mutations) and correlations (e.g., candidate genes from the GWAS Catalog). As of 2025 (version 24.4), the database includes approximately 2 million associations covering 26,000 genes and 39,000 diseases and phenotypes. A key feature of DisGeNET is its data ranking system, which is based on source credibility and validation methods, thereby enhancing the accuracy and reliability of the presented information [62]. The DisGeNET database aggregates data from multiple reputable sources, including UniProt, ClinVar, and Orphanet. Notably, some records in the database, especially those obtained from the GWAS Catalog and the ClinVar archive, may lack direct publication citations (PMIDs), thereby limiting the verifiability of primary studies. Such records without documented confirmation were excluded from analysis in the present paper. DisGeNET is regularly updated in annual releases; each update involves (1) the integration of new data from PubMed, (2) the synchronization with partner databases, and (3) the revision of existing gene–phenotype associations and the recalculation of their reliability scores.

Analysis of the DisGeNET data identified 120 unique biomarkers (Appendix A) associated with ovarian cancer; of those, 115 markers were mentioned for the first time since 2014 (Figure 3c). The association between these biomolecules and the development of ovarian cancer has been confirmed in 2510 academic publications; their number peaked in 2018 (Figure 3b) and then significantly decreased down to 212 publications in 2024. Interestingly, we found no publications in the evidence-based resources contained in DisGeNET that would overlap with our reference list.

A total of 24 potential markers were identified at the intersection of four major data sources: DisGeNET, CancerMine, Open Targets, and our systematic review (Figure 3d). The list included DNA repair regulators (*ATM*, *BRCA1*, *BRCA2*, *PALB2*, *RAD51C*, *MLH1*, *CHEK2*, *BARD1*, *FANCJ*, and *RAD50*), tumor suppressors and cell cycle controllers (*TP53*, *RB1*, *APC*, *PTEN*, and *CDKN2A*), and proteins involved in signaling pathways, cell adhesion, and tumor aggression (*EGFR*, *ERBB2*, *KIT*, *RSPO1*, *MUC16*/*CA125*, *CDH1*, *HNF1B*, *FOS*, and *NF1*). The isolated proteins have been repeatedly validated as molecular markers and targets for ovarian cancer diagnosis and therapy, highlighting their fundamental role and clinical relevance for multi-omics studies of the disease. Appendix A presents the full list of shared biomarkers. The overlapping of markers from the DisGeNET database and the results of our review revealed 34 proteins (Figure 3c) mentioned in publications over the past decade. According to the DisGeNET data and our findings, DNA repair proteins BRCA1 and BRCA2, present in 90% of all the studies, are the most frequently mentioned biomarker proteins (Figure 3e). The frequency of occurrence of the TP53 protein in both the DisGeNET database and our analysis is also high.

Our integrative multi-omics approach led to the identification of 470 genes that, according to DisGeNET, CancerMine, and the Open Targets Platform, have not been previously associated with ovarian cancer. This subset of genes is of particular interest for follow-up studies, as it may include novel contributors to tumor biology, progression, or therapeutic response. Their identification highlights the power of integrative multi-omics analysis in uncovering potential regulatory and functional elements that single-omics or literature-based approaches may overlook.

Enrichment analysis using the Metascape platform [47] revealed the potential regulatory influence of *FOXJ2* on 24 of these 470 genes. *FOXJ2* has been reported in several cancer-related studies [63,64]. In our analysis, *FOXJ2* emerged as a potential upstream regulator of gene sets enriched for cilia-associated and microtubule-based motility pathways, including *DNAI3, DNAI7, CFAP276, CFAP141, TEKT1, ZBBX, PACRG, DNAAF6, LRGUK*, and *MNS1*.

The DisGeNET, CancerMine, and Open Targets Platform databases employ a hybrid approach to data curation, often combining automated methods and expert review. However, these databases embrace not only multi-omics studies but also multiple publications focusing on individual omics levels, which explains the partial discrepancies in biomarker sets.

Meanwhile, only a small percentage of the hundreds of molecules identified have been independently confirmed in several studies, indicating a combination of three key factors. First, multi-omics methods are characterized by low reproducibility. Differences in sample preparation protocols, analytical platforms (such as mass spectrometry and sequencing), and quality control schemes can cause batch effects and artifacts masking true biological signals. Second, the high inter- and intratumoral biological heterogeneity of ovarian cancer tumors results in significant variability of molecular profiles even in patients with the same nosological form. Third, because of analytical differences (normalization, methods used for batch effect correction, and statistical thresholds) and the variety of data analysis tools, identical raw data “yield” different sets of biomarkers.

Furthermore, many biomarkers are found not to be specific for ovarian cancer when a backward search is performed: they are associated with other pathologies (inflammation, endometriosis, and metabolic and neurodegenerative diseases) or are included in the list of 390 proteomic markers of aging reported in the review [65]. This illustrates the “streetlight effect” phenomenon, when researchers primarily identify molecules that are more easily detectable by the technologies used, rather than those characterized by the highest biological specificity.

All these facts demonstrate that a combination of rigorous standardization of laboratory and analytical protocols, multi-site replication, and integration of bioinformatics approaches capable of isolating robust gene–disease associations from variability noise and technical artifacts is needed for identifying truly reliable and specific ovarian cancer biomarkers.

### 3.4. Functional Annotation of Potential Biomarkers for Ovarian Cancer

In the next part of this study, the list of potential biomarker molecules of different types, in gene name format (*n* = 1649), was examined using principal component analysis to identify the main patterns and visualize the results. This approach simplified multidimensional data, made the comparison of different sets of biomarkers more illustrative, and allowed us to develop hypotheses about the potential functional role of the detected genes in the pathogenesis of ovarian cancer. The results are presented in Figure 4a.

We analyzed gene distribution in the principal component space generated with allowance for the integrated features from the UniProt and KEGG databases (Figure 4a). Gene characteristics extracted from the UniProt database annotations (belonging to specific biological processes, molecular functions, cellular components, and keyword strings) and the KEGG database (involvement in pathways) were selected as features. A binary feature vector was generated for each gene, demonstrating in which functions, processes, and pathways this gene is involved according to the databases. The results showed that there is a major set of points on the left and two additional sets on the right and at the top of the diagram, indicating that common molecular mechanisms and specific features of different types of ovarian cancer may exist. Genes associated with other histological types overlap significantly, demonstrating that there is no clear boundary for cancer type identification and explaining the variability in clinical manifestations and responses to therapy.

Next, we compared the number of potential biomarker molecules specific to each tumor type. The largest number of unique molecules (*n* = 900) was observed for HGSC compared to other types. The genetic profile for HGSC, obtained based on the results of Gene Ontology enrichment analysis in the “Biological Process” category (hypergeometric test, Enrichr cutoff = 0.05), reveals the aggressive nature of this subtype. It includes active remodeling of the tumor microenvironment (*p*-value = 1.3 × 10^−5^), cytoskeleton formation (*p*-value = 2.8 × 10^−5^), and intracellular signaling cascades (*p*-value = 9.2 × 10^−4^). For other groups represented by a small number of genes, no accurate functional annotations could be obtained because the subtypes were described in sporadic papers. Such distribution of functional processes helps elucidate the molecular mechanisms of HGSC, where the prominent genome instability enables the targeting of a large number of potential targets [5,7]. For the remaining types, which carry a limited set of unique genes, the current challenge is to study them at the multi-omics level. Our findings highlight the complexity of ovarian cancer oncogenesis and confirm the importance of integrative data analysis for further research and clinical applications.

Figure 4b visualizes the results of analyzing the KEGG pathway for the largest cluster with the predominant content of potential biomarker molecules specific to HGSC. This cluster comprises 1501 molecules and attests to the key signaling and cancer mechanisms involved in the development of this subtype of ovarian cancer. The analysis demonstrated prominent enrichment in pathways such as prostate cancer and pathways in cancer (*p*-value = 4.4 × 10^−18^), which may be indicative of the common BRCA-associated mechanisms of carcinogenesis, including impaired DNA repair [66,67] and activation of embryonic signatures (*OCT4* and *NANOG*) [68,69], which are typical of serous ovarian carcinoma and aggressive subtypes of prostate cancer. The analysis also revealed the PI3K-Akt signaling (*p*-value = 9.1 × 10^−23^) and AGE-RAGE signaling pathways in diabetic complications (*p*-value = 5.6 × 10^−20^), verifying the importance of the activation of these pathways in the pathogenesis of HGSC. The detected gene enrichment in the Fluid shear stress and atherosclerosis pathway (*p*-value = 8.7 × 10^−17^) was unexpected.

A detailed examination of the genes involved in this pathway revealed their association with extracellular matrix remodeling, the regulation of cell adhesion, angiogenesis, and responses to mechanical stimuli. All these processes are critical for invasion, migration, metastatic spread, and the formation of the tumor microenvironment. For this very reason, such a pathway can manifest in the tumorigenic context of high-grade serous carcinoma due to the overlapping molecular mechanisms associated with the remodeling of the vascular system and extracellular matrix, rather than being directly associated with atherosclerosis [70].

Cancer and signaling mechanisms predominate in the range of involved processes; the PI3K-Akt pathway stands out due to its role in maintaining the neoplastic phenotype and is one of the promising targets for targeted therapy [71]. The visually presented pathway categories, shown in different colors, facilitate data systematization and are consistent with the current concept of the multifactorial nature of HGSC, indicating the need for comprehensive approaches to the research and treatment of this disease.

## 4. Conclusions

This review offers a critical evaluation of 51 multi-omics studies on ovarian cancer conducted over the past decade. The analysis reveals a fundamental methodological discrepancy between the conceptual premise of multi-omics research and its actual implementation. While multi-omics approaches are ostensibly designed to integrate diverse molecular layers within the same biological context, the majority of reviewed studies rely on secondary analyses of previously acquired, heterogeneous datasets. This practice diminishes the biological interpretability of the findings and substantially limits their clinical applicability. Although 1649 putative biomarker molecules were identified across multiple omics modalities, only 24 have been independently validated in large-scale reference databases such as DisGeNET, CancerMine, and the Open Targets Platform. The functional annotation of these candidates consistently highlights perturbations in key pathways, including PI3K-Akt signaling, extracellular matrix remodeling, and cell adhesion processes, suggesting convergent molecular mechanisms underlying ovarian tumorigenesis.

At the same time, the findings underscore persistent challenges that hinder the translational potential of multi-omics research. These include the predominance of HGSC as the primary subject of investigation, a lack of standardized experimental and analytical protocols, and the absence of robust frameworks for integrative data analysis. Such limitations compromise reproducibility, obscure biological insight, and curtail the generalizability of results to other histological subtypes of ovarian cancer.

To address these issues, future research must prioritize the generation of multi-omics data from matched biological samples within the same experimental setting, employing harmonized protocols across all omics layers. Greater attention should also be directed toward underrepresented ovarian cancer subtypes to facilitate the identification of subtype-specific biomarkers. Concurrently, there is an urgent need for the development of advanced, scalable computational tools that can integrate heterogeneous, high-dimensional datasets, particularly under the constraints imposed by limited sample sizes. Progress in this domain will additionally require community-level efforts aimed at standardizing methodological practices, establishing benchmarking resources, and fostering transparent data sharing.

In conclusion, while multi-omics approaches hold considerable promise for advancing biomarker discovery, patient stratification, and therapeutic innovation in ovarian cancer, their current application remains fragmented and suboptimal. The realization of their full potential will necessitate rigorously designed studies that align molecular data generation with clinical context, supported by unified methodological standards and integrative analytical frameworks.

## Figures and Tables

**Figure 1 ijms-26-05961-f001:**
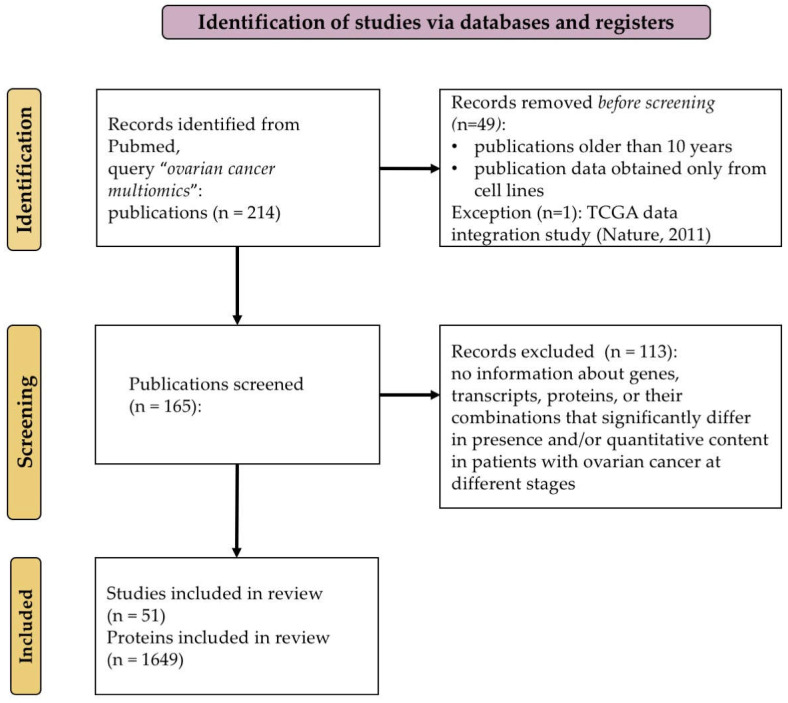
PRISMA [33] flow diagram of the literature screening and selection processes.

**Figure 2 ijms-26-05961-f002:**
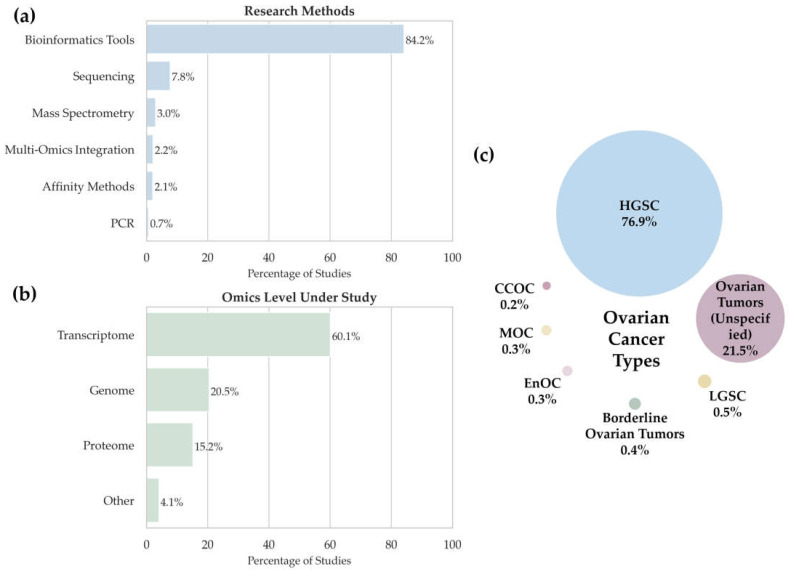
The distribution of studies according to (**a**) research methods used in the multi-omics studies, (**b**) types of omics data, and (**c**) types of ovarian cancer in the study sample (high-grade serous carcinoma (HGSC); ovarian tumors, unspecified; low-grade serous carcinoma (LGSC); clear cell ovarian carcinoma (CCOC); borderline ovarian tumors; mucinous ovarian cancer (MOC)).

**Figure 3 ijms-26-05961-f003:**
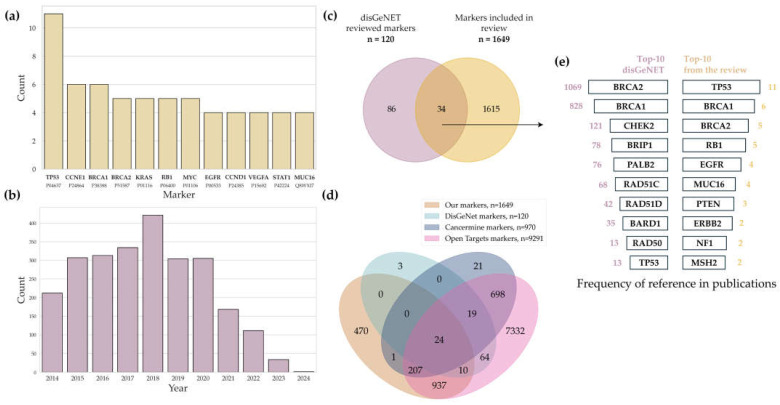
(**a**) The most common protein biomarkers included in our systematic review. Y axis (Count) denotes the number of publications mentioning each biomarker; (**b**) temporal trends of publications about ovarian cancer biomarkers in the DisGeNET database; (**c**) the overlap of biomarkers presented in the DisGeNET database and those identified in our systematic review; (**d**) comparison of potential markers of four major data sources: DisGeNET, CancerMine, Open Targets, and our systematic review; (**e**) comparative frequencies of the top 10 biomarkers among the 34 molecules simultaneously found in both the DisGeNET database and this review (numbers next to the bars denote the number of publications).

**Figure 4 ijms-26-05961-f004:**
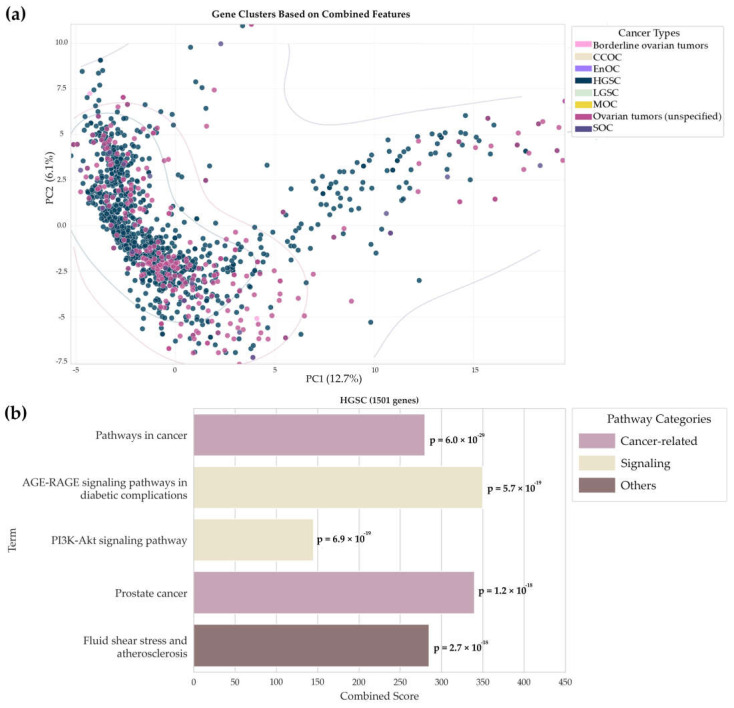
(**a**) The distribution of genes in the principal component space generated with allowance for integrated features from the UniProt and KEGG databases, where each point corresponds to a separate gene and color indicates the histological type of ovarian cancer. EnOC—endometrioid ovarian carcinoma; HGSC—high-grade serous ovarian carcinoma; SOC—serous ovarian cystadenocarcinoma; (**b**) KEGG pathway enrichment analysis for identifying the most important signaling and metabolic pathways characteristic of genes in the cluster predominantly containing HGSC-associated genes. The combined score is calculated by multiplying the log of the *p*-value obtained using Fisher’s exact test by the z-score, which measures the deviation from the expected rank. This allows for the estimation of the significance and rank of enriched pathways.

## Data Availability

All authors confirm that all data and materials support their published claims and comply with field standards and are included in this article and the Appendix A.

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
