# Peer review of "Ovarian Cancer: Multi-Omics Data Integration"

_ijms, 2025, doi:10.3390/ijms26135961_

Round 1

Reviewer 1 Report

Comments and Suggestions for Authors

Context: Ovarian cancer is a highly heterogeneous and aggressive gynecological malignancy, which is often diagnosed at an advanced stage when curative treatments are limited. Despite significant advances in the treatment of ovarian cancer, the overall 5-year survival rate remains below 45% due to late diagnosis, high relapse rates, and development of chemo-resistance. Recent technological advances in multi-omics approaches have provided new insight into the complex molecular landscape of ovarian cancer. However, the integration of these large-scale datasets remains methodologically challenging and poorly standardized.

Results: The authors therefore conducted a systematic review and integrative analysis of 51 multi-omics studies on ovarian cancer to identify recurrent protein biomarkers and methodological variability across studies. The authors compiled a comprehensive list of 1,649 potential protein biomarkers that were extracted from genomic, transcriptomic, proteomic, and epigenomic datasets. Among the existing ovarian cancer subtypes, high-grade serous carcinoma (HGSC) emerged as the most frequently studied, representing more than 75% of the analyzed cases.

Functional annotation revealed that commonly reported biomarkers such as p53, BRCA1/2, Cyclin E1, and Myc are involved in key oncogenic processes including DNA repair, cell cycle regulation, and immune evasion. Principal component and KEGG pathway enrichment analyses showed that these biomarkers are enriched in signaling pathways such as PI3K-Akt, AGE-RAGE, and fluid shear stress response, underscoring their potential roles in tumor progression and metastasis.

A comparative analysis with the DisGeNET database identified limited overlap (~4%) between literature-derived and database-listed biomarkers, highlighting issues of reproducibility and methodological heterogeneity.

The authors identified a series of key challenges that contribute to the limited reproducibility and comparability of biomarker findings across studies, including substantial variation in sample preparation protocols, differences in data acquisition platforms, and inconsistencies in bioinformatics pipelines, such as normalization methods and statistical thresholds. 

The authors conclude that significant limitations in study design, data integration methods, and cross-study comparability must be addressed to enable the identification and validation of clinically reliable biomarkers for ovarian cancer using multi-omics approaches.

Critique:

  1. Overall structure of the manuscript: Although the topic of this manuscript is timely and of great interest to the field, the presentation of the manuscript largely undermines its clarity and impact. The introduction is not well-structured which limits the understanding of the rationale of the study and its objectives. The authors also scatter basic knowledge about ovarian cancer, definitions of multi-omics and technical difficulties and previous literature with no smooth narration. Consequently, the reader remains unsure as to what gap the review aims to address, or what sets it apart from existing works. The results section also lacks focus, mixing raw data, a somewhat scattered functional interpretation and speculative discussion aspects that would be better left to a separate discussion section. Finally, the materials and methods section lacks essential details regarding how the data were analyzed and integrated. For instance, the criteria used for study inclusion, the specific methods for extracting and annotating biomarkers, and the bioinformatic tools or parameters applied (are either vaguely described or entirely omitted. This lack of methodological transparency raises concerns about the reproducibility and reliability of the findings.
  1. Structural and methodological limitations of the present study: The authors report that 84.2% of the studies relied on bioinformatic tools, suggesting a dominance of in silico approaches in the discovery of biomarkers relevant to ovarian cancer. However, this categorization is conceptually problematic, as most of the other methodological approaches listed also inherently depend on bioinformatic tools for data processing, analysis, and interpretation. It is therefore unclear how the authors defined and separated “bioinformatic tools” as a stand-alone category.
  2. Imbalance in the representation of the different ovarian cancer subtypes: Another major caveat of this analysis is the marked overrepresentation of data derived from HGSC, which accounts for over 75% of the included studies. While HGSC is indeed the most prevalent and subtype of ovarian cancer, this disproportionate focus significantly limits the generalizability of the findings to other histological subtypes. This limitation should be explicitly acknowledged by the authors throughout the manuscript.
  1. Limited novel insight from the integration of multi-omics data: the authors highlight several genes, including TP53, BRCA1/2, MYC, CCNE1, and KRAS, as top biomarkers recurrently identified across studies. While these genes are central to the pathogenesis of ovarian cancer, their strong association with the disease is already well-established in the literature and clinical practice. The manuscript does not clearly distinguish whether any novel or less-characterized biomarkers emerged from this integrative analysis. This represents a missed opportunity, as one of the promises of multi-omics approaches is the ability to identify previously unrecognized molecular players.

In summary, this manuscript compiles a substantial number of studies and provides a potentially valuable resource for the ovarian cancer field. However, the lack of structural disorganization, the methodological ambiguity, the poor emphasis on novel findings, and the overrepresentation of a single subtype of ovarian cancer significantly limit its overall impact. The manuscript would benefit from a clearer articulation of its objectives, a more structured distinction between descriptive results and interpretative commentary, enhanced methodological transparency, and a more critical, insight-driven synthesis of the existing literature.

Comments on the Quality of English Language

The quality of the English language is generally acceptable but would benefit from careful editing for clarity, grammar, and conciseness. Several sentences are overly long or complex, making them difficult to follow. In some instances, imprecise wording or awkward phrasing weakens the impact of key points.

Author Response

We thank the reviewer for the time and effort spent revising the manuscript. We have adjusted the text in accordance with the comments provided, and in our opinion, this has significantly improved the article. 

  1. Overall structure of the manuscript: 

We thank the reviewer for this valuable contribution to our work. All comments have been carefully checked and taken into account in our manuscript. We have structured the narrative more clearly, making smoother transitions between sections. Also, the article format has been changed to "systematic review", the "materials and methods" section has been moved after the introduction, which makes it easier to understand the work and how the data were analyzed and integrated. The Discussion section has been renamed to "Conclusions" to highlight speculative discussion aspects.

  1. Structural and methodological limitations of the present study:

We thank the reviewer for this insightful comment. We agree that most modern molecular studies, including those involving sequencing and proteomics, inherently rely on bioinformatic tools for data processing and interpretation. In our classification, we used the category “bioinformatics tools” to denote studies based primarily on secondary analysis of publicly available omics datasets (e.g., TCGA, GEO), without new experimental data generation. These in silico studies often focus on computational modeling, network analysis, and biomarker discovery using existing data resources. By contrast, other categories—such as sequencing or mass spectrometry—referred to studies involving primary data acquisition, even though bioinformatic analysis was integral to them as well.

To avoid misunderstanding, we have clarified this distinction in the revised text of Section 3.1.1.

  1. Imbalance in the representation of the different ovarian cancer subtypes

We thank the reviewer for the valuable comment. The sentence “...while HGSC is indeed the most prevalent subtype of ovarian cancer, this disproportionate focus significantly limits the generalizability of the findings to other histological subtypes” was inserted into the section “Integration of Multi-Omics Data and Their Functional Characterization.” Text “The predominance of high-grade serous carcinoma in research, along with the lack of unified protocols and integration strategies, significantly limits generalizability across histological subtypes” was added  in the “Conclusions” section.

  1. Limited novel insight from the integration of multi-omics data: 

Thank you for your insightful feedback regarding our manuscript. We appreciate your emphasis on the importance of identifying novel or less-characterized biomarkers through multi-omics integration. We performed an additional analysis to systematically evaluate the novelty of the molecular features identified in our study.

Specifically, we cross-referenced the full set of prioritized genes against three independent cancer-related knowledge bases (CancerMine, Open Targets Platform, and DisGeNET) in order to assess prior associations with cancer phenotypes. This analysis revealed a subset of 470 genes that are not currently annotated as known oncogenes, tumor suppressors, or cancer-associated targets in these resources.

To further explore the relevance of these genes, we performed transcription factor enrichment and functional clustering analyses using the Metascape platform. Among the 470 genes, we identified 24 genes that are potential targets of FOXJ2, a transcription factor whose role in ovarian cancer pathogenesis was described just few months ago [https://pmc.ncbi.nlm.nih.gov/articles/PMC11881463/]. These targets were significantly enriched for cilia-associated and microtubule-based motility pathways, suggesting potential biological relevance in tumor cell motility and signaling. This finding illustrates how our integrative approach can uncover underexplored gene networks that may contribute to cancer biology, even when they are not yet well documented in literature-based resources.

We have revised the manuscript to include this new analysis to the results and  discussion.

Comments on the Quality of English Language

Thank you for pointing this out. We have carried out the required careful editing for clarity, grammar, and conciseness. Additionally, our manuscript has been checked for inaccurate wording and long sentences, and it has been revised in accordance with the recommendations.

Reviewer 2 Report

Comments and Suggestions for Authors

The study provide comprehensive review of multiomics approach for identifying biomarker. The methodology is written well. The authors have well justified all the statistical methods they used to conclude the results. In general introduction provide enough content to build the hypothesis. Results and discussion is also presented well with deeply analyzing results as well comparing with other database information. I have following recommendations to improve the quality-

1) The subheading 2.1 has pretty good text on results of multiomics studies in Ovarian cancer. It is difficult to read it in one flow. So, I would recommend to further split into subheading to make it more readbale. Fore example, Transcriptomics part can be separated from proteomic part. Similarly for mutations and epigenetic alterations. 
2) Do you know why there were very high OV studies in year 2018? After that its decreasing trend. any specific reason for this trend?
3) Because the study focused on multiomics integration, I would recommend adding a paragraph describing how people have used different approach to integrate multiomics data for predicting subtype or biomakrers. Fig 2a, demonstrated an overview aspect of techniques used in multiomics analysis; however this needs to be more comprehensive. For example, PINSPlus (https://pubmed.ncbi.nlm.nih.gov/30590381/), CIMLR (https://pubmed.ncbi.nlm.nih.gov/30367051/), iCluF (https://pubmed.ncbi.nlm.nih.gov/38698887/), Subtype-GAIN (https://pubmed.ncbi.nlm.nih.gov/33599254/),  etc. used different approaches for integrating multiomics data for specific biological objectives in cancer including Ovarian cancer. This para can more methodological, but informatics in the current context. 
4) I think its a good strategy to compare results with DisGeNET, however there low overlaps for biomarkers. Any specific reason?
5) Line 324-325: Is it a heading ? Comparative Analysis of Ovarian Cancer Biomarkers According to the Literature 324 Data and the DisGeNET Database. Make it clear. There are other similar issues in the manuscript. 
6) Figure 4a, Please add the % component for both PCA1 and 2. That way, reliability can be assessed more.    
7) What is combined score in Figure 4b? How was it calculated? briefly explain in the text.

Author Response

We appreciate the reviewer for their time and effort in revising the manuscript. We have made adjustments to the text based on the provided comments, and we believe this has greatly enhanced the quality of the article.

1) Thank you for your thoughtful comments regarding the stylistic complexity of the text and the structure of section 2.1. 

In response, Section 2.1 (now 3.1) was restructured into three subsections corresponding to the analytical dimensions of Figure 2: 3.1.1 Analytical Methods, 3.1.2 Omics Data Types, and 3.1.3 Ovarian Cancer Subtypes. Within subsection 3.1.2, we now clearly distinguish between transcriptomic, genomic, proteomic, and other omics approaches, including a mention of mutation profiling and epigenetic regulation.

While we did not assign individual sub-subheadings to each omics level to preserve narrative continuity, we aimed to clarify the discussion by grouping and thematically separating the main data types. We hope this reorganization improves the section’s readability while maintaining its integrative character.

2) Thank you for the interesting question! We can assume that this is due to the active use and reanalysis of The Cancer Genome Atlas (TCGA) data. In 2018, the formation, structuring, and integration of ovarian cancer data in the GDC (The National Cancer Institute’s Genomic Data Commons) were completed, which contributed to a sharp increase in in silico publications

3) We sincerely appreciate the reviewer's valuable suggestion regarding the need to strengthen our discussion of multiomics integration methods. In response to this comment, we have expanded our methodological discussion by adding text to the paragraph beginning with "The development of computational algorithms" in the "Introduction" section that systematically reviews key computational approaches for multiomics data integration in ovarian cancer research. This addition specifically examines the referenced methods (PINSPlus, CIMLR, iCluF, and Subtype-GAIN) and their applications in cancer subtyping and biomarker discovery.

4) DisGeNET integrates over 400,000 associations, with approximately 60% derived through automated text mining of scientific literature. This approach enables rapid and large-scale extraction of genotype–phenotype associations, addressing the growing challenge of manually curating such information in a structured format [https://academic.oup.com/nar/article/48/D1/D845/5611674].

However, this strategy also introduces limitations that may explain the observed low overlap in biomarker associations. The reliance on automated text mining can lead to decreased precision, especially in complex or ambiguous contexts. In our analysis, we identified several gene-disease associations in DisGeNET that were clearly erroneous—likely due to issues such as misinterpretation of negations, co-occurrence biases, or lack of contextual understanding by the text-mining algorithms. We have reported these findings to the DisGeNET team.

In contrast, our approach combines automated extraction techniques with manual curation and expert validation. This hybrid methodology allows us to take advantage of the scalability and efficiency of text mining while maintaining a higher level of precision through human oversight. The observed differences in results between our study and DisGeNET may, in part, reflect these methodological distinctions, which is mentioned in our work. We believe that by employing both approaches, researchers and readers are presented with a more comprehensive and balanced view of the biomarker landscape, which strengthens the interpretability and reliability of the findings across different contexts.

5) We thank the reviewer for his valuable comment. Headings in the text have been checked and highlighted in bold. 

6) We have now added the percentage of explained variance for each principal component to Figure 4a (PC1: 12.7%, PC2: 6.1%) as suggested. This addition provides further clarity on the contribution of each component to the total variance.

7)  We thank the reviewer for the comment. The clarification has been made in the captions to figure 4.

The combined score is calculated by multiplying the log of the p-value obtained using Fisher's exact test by the z-score, which is a measure of deviation from the expected rank. This allows for the estimation of the significance and rank of enriched pathways. 

Round 2

Reviewer 1 Report

Comments and Suggestions for Authors

The authors have carefully addressed the concerns raised during the initial review, which has significantly improved the structure, clarity, and overall quality of the manuscript. In particular, the organization of the results and discussion is now more coherent, and the use of figures and supplementary materials enhances the accessibility of the findings. However, the conclusion section could benefit from further improvement. At present, it reiterates many points already discussed without clearly summarizing the novel insights of the study or articulating how the identified limitations could be addressed in future research. A more structured conclusion that succinctly highlights the key contributions, outlines practical implications, and proposes clear next steps would strengthen the impact of the manuscript and provide better closure.

Comments on the Quality of English Language

The manuscript is generally well written, with clear and coherent language throughout. The authors use appropriate scientific terminology, and the narrative is easy to follow. However, minor grammatical and stylistic improvements, particularly in the conclusion section, would enhance overall readability.

Author Response

We would like to express our sincere gratitude to the reviewer for their thoughtful feedback on our manuscript, which has greatly enhanced the quality of our work. We have meticulously reviewed the entire text to address minor grammatical and stylistic issues. Additionally, we have revised the conclusion section for clarity and added graphical abstract to further illustrate our findings. We believe these changes have significantly improved our manuscript.

The changes in the text highlighted in yellow and the rewritten conclusion section are attached to this reply.

Reviewer 2 Report

Comments and Suggestions for Authors

All the comments were addressed. 

Author Response

We thank the reviewer for the high evaluation of our work. We hope that, after the second round of corrections, our work has significantly improved.